# Current State-of-the-Art Therapy for Malignant Pleural Mesothelioma and Future Options Centered on Immunotherapy

**DOI:** 10.3390/cancers15245787

**Published:** 2023-12-10

**Authors:** Susana Cedres, Augusto Valdivia, Patricia Iranzo, Ana Callejo, Nuria Pardo, Alejandro Navarro, Alex Martinez-Marti, Juan David Assaf-Pastrana, Enriqueta Felip, Pilar Garrido

**Affiliations:** 1Medical Oncology Department, Vall d´Hebron Institute of Oncology (VHIO), Vall d’Hebron Hospital Universitari, 08035 Barcelona, Spain; avaldivia@vhio.net (A.V.); piranzo@vhio.net (P.I.); acallejo@vhio.net (A.C.); npardo@vhio.net (N.P.); anavarro@vhio.net (A.N.); amartinezmarti@vhio.net (A.M.-M.); jassaf@vhio.net (J.D.A.-P.); efelip@vhio.net (E.F.); 2Thoracic Cancers Translational Genomics Unit, Medical Oncology Department, Vall d´Hebron Institute of Oncology (VHIO), Vall d´Hebron Hospital Universitari, 08035 Barcelona, Spain; 3Medical Oncology Department, Ramón y Cajal University Hospital, 28034 Madrid, Spain; pilargarrido@gmail.com

**Keywords:** malignant pleural mesothelioma, immunotherapy, biomarkers

## Abstract

**Simple Summary:**

Advances in malignant pleural mesothelioma research have led to the approval of first-line immunotherapies and the development of numerous trials of new first-line treatment regimens. Knowledge of predictive factors of response to treatments could help identify patients who benefit from treatments. The purpose of this review is to describe the state of the art in the treatment of patients with MPM at present.

**Abstract:**

Malignant pleural mesothelioma (MPM) is a locally aggressive disease related to asbestos exposure with a median survival for untreated patients of 4–8 months. The combination of chemotherapy based on platinum and antifolate is the standard treatment, and the addition of bevacizumab adds two months to median survival. Recently, in first-line treatment, immunotherapy combining nivolumab with ipilimumab has been shown to be superior to chemotherapy in the CheckMate-743 study in terms of overall survival (18.1 months), leading to its approval by the FDA and EMA. The positive results of this study represent a new standard of treatment for patients with MPM; however, not all patients will benefit from immunotherapy treatment. In an effort to improve the selection of patient candidates for immunotherapy for different tumors, biomarkers that have been associated with a greater possibility of response to treatment have been described. MPM is a type of tumor with low mutational load and neo-antigens, making it a relatively non-immunogenic tumor for T cells and possibly less susceptible to responding to immunotherapy. Different retrospective studies have shown that PD-L1 expression occurs in 20–40% of patients and is associated with a poor prognosis; however, the predictive value of PD-L1 in response to immunotherapy has not been confirmed. The purpose of this work is to review the state of the art of MPM treatment in the year 2023, focusing on the efficacy results of first-line or subsequent immunotherapy studies on patients with MPM and possible chemo-immunotherapy combination strategies. Additionally, potential biomarkers of response to immunotherapy will be reviewed, such as histology, PD-L1, lymphocyte populations, and TMB.

## 1. Introduction

Malignant pleural mesothelioma (MPM) is a rare, highly lethal cancer associated with asbestos exposure, with peak incidence expected in the 2020s in developed countries [1]. MPM can be classified according to histology from best to worst prognosis as epithelioid, biphasic, and sarcomatoid. The prognosis of patients with MPM is poor, with a median survival of 9–18 months.

For many years, chemotherapy based on platinum plus pemetrexed has been the standard treatment for patients with MPM, but recently, the combination of nivolumab and ipilimumab showed improvements in overall survival compared to chemotherapy [2,3]. After the first-line treatment, there are no other options for systemic treatment approved for MPM.

Genomic studies have shown that mesothelioma is mainly defined by the loss of function of tumor suppressor genes, with mutations in *BAP1*, *CDKN2A*, and *NF2* being the most common alterations [4,5]. The chronic inflammatory response to asbestos creates a unique tumor environment composed primarily of immunosuppressive cells (regulatory T cells, macrophages, and myeloid-derived suppressor cells) [6]. MPM expresses multiple immune checkpoint inhibitors. VISTA expression in MPM is much higher compared to other tumors and is only expressed in tumors of the epithelioid subtype [5]. PD-L1 expression is positive in approximately 40% of patients [7,8,9,10]; it is more common in the non-epithelioid subtype and is associated with a worse prognosis [7,8,9,10,11].

In this review, we provide an overview of current therapeutic options for patients with MPM and discuss potential biomarkers of response to immunotherapy and new future options. 

## 2. Treatment

### 2.1. Surgery

Most patients with mesothelioma are not candidates for surgery due to the extent of the disease and comorbidities. The MARS study, which included 50 patients, demonstrated that extrapleural pneumonectomy was associated with severe complications and did not improve survival compared to patients who did not undergo surgery [12]. In 2023, the results of the MARS-2 study were presented, comparing the treatment of chemotherapy and pleuro-decortication versus chemotherapy without surgery [13]. The study found that survival was superior for patients who did not undergo surgery. However, this study has not yet been published, and the results should be interpreted with caution. More patients with tumors spreading to the lungs were included in the surgery arm. In addition, the preoperative staging of patients was not standardized, and there were no volumetry pre- and post-chemotherapy treatments for the evaluation of therapeutic response according to the modified RECIST criteria. These issues need to be rectified to establish the role of surgery in MPM. 

### 2.2. First Line

Regarding systemic treatment, the standard treatment for almost two decades has been antifolate and platinum chemotherapy [2]. The pivotal EMPHACIS trial demonstrated that the combination of cisplatin and pemetrexed induces an increase in survival (OS) of 9.3 to 12.1 months compared to cisplatin alone. In the MAPS study, the addition of bevacizumab to chemotherapy with cisplatin and pemetrexed led to an increase in OS from 16.1 to 18.8 months and in progression-free survival (PFS) from 7.3 to 9.2 months [14]. However, bevacizumab is not approved by the FDA or EMA for treating MPM.

After almost two decades without new, relevant advances in first-line treatment for MPM, in May 2019, the FDA approved the NovoTTF therapy (alternating electric field therapy, TTFields) in combination with platinum pemetrexed [15]. The phase II STELLAR trial resulted in an OS of 18.2 months and a PFS of 7.6 months, with both results being favorable compared to historical cohorts.

The third regulatory approval of first-line treatments took place in 2020 with the combination of nivolumab plus ipilimumab (reviewed later) [3]. The results of this study have marked a change in the standard treatment of mesothelioma and opened the doors to new research studies on other immunotherapy molecules for use in the first line of treatment.

### 2.3. Second Line

There are currently no approved therapies for when patients progress to first-line platinum-based treatment. The most widely adopted treatment options include single-agent chemotherapy with vinorelbine or gemcitabine or retreatment with pemetrexed. These options are primarily based on retrospective or phase II studies reporting an overall response rate (ORR) of 0–19%, a disease control rate of 38–84%, and a PFS of 1.6–3.8 months, with an OS ranging from 2.5 to 12 months [16]. The results of the RAMES trial, the study with the largest number of second-line patients, have recently been published, comparing the combination of ramucirumab and gemcitabine versus gemcitabine and demonstrating an improvement in OS for the combination [17].

## 3. Immunotherapy

### 3.1. Second and Futher Lines

The first studies with anti-CTLA-4 yielded negative results. Tremelimumab was evaluated in the phase 2b DETERMINE study in a large cohort of patients versus a placebo and did not demonstrate an improvement in OS (7.7 and 7.3 months for patients treated with tremelimumab and the placebo, respectively) [18].

Subsequently, anti-PD-1 demonstrated promising results in phase II studies, but they were not confirmed in all phase III studies (Table 1). Pembrolizumab achieved a disease control rate of 72% in the phase Ib KEYNOTE-028 study [19]. Unfortunately, these good results were not confirmed in the randomized PROMISE-meso study that compared pembrolizumab with chemotherapy among patients who progressed to platinum and achieved a PFS of 2.5 months with pembrolizumab versus 3.4 months with chemotherapy [20].

However, with nivolumab, an anti-PD-1 monoclonal antibody, the preliminary results of activity were confirmed in a phase III trial. The phase II NivoMes and MERIT studies demonstrated a response rate of 24–29% and OSs of 11.8 and 17.3 months [21,22]. Based on the results of this study, nivolumab was approved by the Japanese health authorities in August 2018 for patients who progress to chemotherapy. Subsequently, the phase III CONFIRM trial evaluated nivolumab versus a placebo in the second or third line [23]. Most patients had epithelioid histology (88%) and were PD-L1-negative (66%). The study met its primary objectives, with a median PFS of 3.0 months for nivolumab versus 1.8 months for the placebo and a median OS of 9.2 with nivolumab and 6.6 months with the placebo. The results of this study have not led to the approval of nivolumab in the second line of treatment.

Finally, avelumab, an anti-PD-L1 monoclonal antibody, was explored in a phase 1b study, achieving a response rate of 9%, a PFS of 4.1 months, and an OS of 10.7 months [24].

Regarding combinations of immunotherapy drugs, several studies have published the results of the combination of anti-PD-1 therapy with anti-CTLA-4. The NIBIT-MESO-1 study was a phase II, open-label, non-randomized trial involving patients with pleural or peritoneal mesothelioma in the second or third line that were administered tremelimumab in combination with durvalumab [25]. The study achieved a response rate of 28% with a duration of response of 16.6 months, a PFS of 5.7 months, and a median OS of 16.5 months. In a recent 4-year study update, it was reported that the combination of tremelimumab with durvalumab was associated with durable survival (20% at 36 months and 15% at 48 months) [26].

Two studies have explored the combination of nivolumab with ipilimumab for treating patients who relapse after the first line of treatment: the INITIATE study and the MAPS-2 [27,28]. The studies reported a response rate of around 30%, a PFS of 6 months, and an OS of 15.9 months.

### 3.2. Immunotherapy in First Line

CheckMate-743 is an open-label, randomized, phase III study comparing first-line treatment with nivolumab plus ipilimumab versus chemotherapy for patients with previously untreated unresectable MPM with an ECOG score of 1 or less [3]. A total of 605 patients were included, who were randomized 1:1 to receive nivolumab at a dose of 3 mg/kg every 2 weeks with ipilimumab 1 mg/kg every 6 weeks for two years versus chemotherapy with platinum and pemetrexed. The results of the study were positive, demonstrating a 4-month survival advantage for the immunotherapy combination (18.1 vs. 14.1 months). The 3-year survival rate was 23.2% for immunotherapy versus 15.4% for chemotherapy [29]. The median PFS was 6.8 months with nivolumab plus ipilimumab versus 7.2 months with chemotherapy, and there was no difference in the response rate (40% and 43% for immunotherapy and chemotherapy).

In terms of toxicity, grade 3–4 adverse events were experienced by 31% of patients treated with immunotherapy and 32% of patients treated with chemotherapy. However, for patients who discontinued immunotherapy due to toxicity, survival was not affected, reaching a median OS of 25.4 months from randomization.

The magnitude of the survival benefit with nivolumab plus ipilimumab was greater for patients whose PD-L1 expression was ≥1%, but it should be noted that PD-L1 expression was not a stratification factor in this study.

In a prespecified histology-based exploratory analysis, subgroups of patients with epithelioid histology had an OS of 18.7 months with immunotherapy versus 16.2 months with chemotherapy (HR 0.85). However, in the subgroup of patients with non-epithelioid histology, the differences were greater, with an OS of 18.1 months versus 8.8 months for immunotherapy versus chemotherapy, respectively (HR 0.46). Although the benefit of immunotherapy seems greater with respect to non-epithelioid tumors, after adjusting for treatment, it was observed that the OSs of the patients treated with immunotherapy in the group of epithelioid and non-epithelioid tumors were similar (with median OSs of 18.7 months and 18.1 months for the epithelioid and non-epithelioid patients). It is postulated that non-epithelioid tumors have a worse prognosis and are less sensitive to chemotherapy treatment.

An exploratory biomarker analysis included a four-gene gene expression signature (*CD8A*, *STAT1*, *LAG3*, and *CD27*), TMB (tumor mutation burden), and the pulmonary immune prognostic index (LIPI) measured according to LDH levels and neutrophil/lymphocyte ratios (NLR) in peripheral blood [29]. A positive correlation was detected in patients with high inflammatory gene signature expression with a survival benefit from immunotherapy (21.8 months for patients with a high inflammatory signature versus 16.8 months for patients with low scores). Of the other two markers studied, neither the TMB nor the LIPI score were predictive of survival.

Regarding the analysis of quality of life, immunotherapy imparted an improvement in symptoms accompanied by the maintenance of general condition and a reduction in the risk of definitive deterioration of symptoms related to the disease during treatment [30].

The results of this study led to the approval by regulatory agencies (FDA and EMA) of the combination of nivolumab plus ipilimumab as a first-line treatment for unresectable MPM.

### 3.3. Combination of Immunotherapy with Chemotherapy

Prior to the publication of the CheckMate-743 study, the results of two phase II studies that analyzed the effectiveness of the combination of immunotherapy with durvalumab added to the first line of chemotherapy were presented [31,32]. The DREAM study included 54 patients who received durvalumab in combination with cisplatin and pemetrexed for a total of six cycles followed by durvalumab until the completion of one year or progression [31]. The trial exceeded its prespecified objective, attaining a 6-month PFS rate of 57%. The median PFS was 7 months, and OS was 18.4 months. In a post hoc analysis, responses were observed in all histological types, and no significant association was detected between PD-L1 expression and PFS. The same treatment scheme was used in the PrE0505 study, finding a response rate of 56.4% and a PFS of 6.7 months [32]. The median OS of 20.4 months was significantly longer than that of the historical control. Regarding analysis according to histology, patients with epithelioid tumors had a higher response rate than patients with non-epithelioid tumors (65.9% versus 28.6%, *p* = 0.03). Based on these results, a randomized phase III study (DREAM3R) was conducted to compare whether there are survival differences with the addition of durvalumab to standard platinum treatment, the results of which are pending.

The third study of combination chemotherapy plus immunotherapy in the first line is the phase 2 JME-01 study [33]. This was a single-arm treatment study combining cisplatin with pemetrexed and nivolumab for 4–6 cycles, followed by a maintenance phase with nivolumab until progression or reaching unacceptable toxicity levels. The objective response rate was 77%, including responses for all histologies, with a duration of response of 6.7 months, a PFS of 8 months, and an OS of 20.8 months.

During the year 2023, the results of the first phase III study of chemotherapy with immunotherapy in the first line were presented [34]. The IND.227 study was a phase 2–3 study that evaluated the addition of pembrolizumab to chemotherapy. In phase II, the trial did not demonstrate differences in PFS (6.7 and 6.8 months), but it did reveal differences in OS (19.8 versus 8.9 months for combination and chemotherapy respectively) [35]. Despite not meeting the primary endpoint of PFS, numerically superior survival data for the combination with pembrolizumab led to the completion of enrollment. In phase III, 440 patients were included and stratified by histology. The study demonstrated improved OS for the chemo-immunotherapy combination, with a median OS of 17.3 months versus 16.1 months in favor of the pembrolizumab group. Three-year survival was 25% vs. 17% for immunotherapy. However, as in the CheckMate-743 study, no differences were observed in PFS (7.1 months in both treatment groups), and a greater survival benefit was observed for patients with non-epithelioid histology (with an OS of 12.3 versus 8.1 months in favor of pembrolizumab for non-epithelioid tumors and 19.8 vs. 18.2 months for epithelioid tumors). No significant differences were observed according to the level of PD-L1 expression.

## 4. Predictive Factors of Response to Immunotherapy

The identification of biomarkers that allow us to determine which patients with MPM may respond to immunotherapy could improve the cost/benefit ratio and potentially facilitate the rational development of novel combination strategies for overcoming primary resistance.

### 4.1. PD-L1

The predictive role of PD-L1 in immunotherapy response in MPM is unclear (Table 2). Early phase I and II studies have found conflicting results regarding a higher response rate for PD-L1-positive tumors treated with immunotherapy compared to negative ones. However, in the phase 2 KEYNOTE-158 study and in the phase III PROMISE study, no differences were demonstrated in PD-L1-expressing patients compared to PD-L1 negative patients [20,36].

Regarding nivolumab, four trials have shown contradictory results. In the NivoMes trial, no differences in response or survival were demonstrated when stratifying patients by PD-L1 status [21]. In contrast, in the Japanese MERIT trial, which tested nivolumab in a similar pretreated population, an interesting (though not statistically significant) trend was reported in favor of PD-L1 positivity compared to PD-L1 negativity in terms of response rate, PFS, and OS [22]. Finally, in the randomized phase 3 CONFIRM study, PD-L1 expression ≥1% was not related to survival [23].

Results regarding the combination of nivolumab with ipilimumab have indicated that there is a modest association between PD-L1 protein expression and outcomes. In the INITIATE trial, a post hoc analysis of 12-week response and duration of response according to PD-L1 status suggested a greater benefit for PD-L1-positive compared to negative tumors [28]. The MAPS2 trial also reported an advantage in terms of response rate but not in terms of duration of response at 12 weeks for patients with PD-L1-positive tumors [27]. In the CheckMate-743 trial, in which patients were not stratified by PD-L1 status, PD-L1 positivity appeared to predict better outcomes with nivolumab plus ipilimumab compared to chemotherapy [3]. Patients with tumors expressing PD-L1 at < 1% had an OS of 17.3 months, which can be compared to 18 months for PD-L1 ≥ 1% patients.

When checkpoint inhibitors are combined with chemotherapy, the predictive value of PD-L1 for response to treatment is even lower. Neither the DREAM trial nor IND227 found any association with OS or PFS with PD-L1 > 1% [31,37].

Finally, in a meta-analysis of 29 publications, PD-L1-positive patients who did not receive immunotherapy appeared to have a worse prognosis compared to PD-L1-negative patients [38]. In contrast, among patients who received immunotherapy, similar or better survival was observed among PD-L1-positive individuals. These findings suggest that the initial prognosis of patients with PD-L1-positive tumors may be worse than that of patients with PD-L1-negative tumors and that PD-L1-targeted therapies may improve survival.

### 4.2. Histology

In the CheckMate-743 study, a difference in efficacy according to histology was reported [3]. In this study, patients were stratified by histology, and 76% of the included patients presented epithelioid histology. Immunotherapy showed a greater benefit than chemotherapy for patients with non-epithelioid as opposed to epithelioid histology, with median OSs of 18.1 months in the immunotherapy arm and 8.8 months in the chemotherapy arm (HR 0.46, 95% CI: 0.31–0.68) for non-epithelioid histology compared with 18.7 months and 16.5 months for epithelioid histology (HR 0.86, 95% CI 0.69–1.08), respectively. The difference in the benefit of immunotherapy according to histology seems to be mainly related to the potential lower efficacy of chemotherapy in the non-epithelioid subtype since the median OS with immunotherapy was the same for all histology types (18 months). It should be considered that this trial was not specifically designed to identify a difference according to histological subtype.

Identical results were found in the IND227 study that evaluated the addition of pembrolizumab to first-line chemotherapy [34]. In this study, patients were stratified by histology, and 78% of patients had epithelioid histology. As in the checkmate 743 study, the difference in survival was greater for the patients with non-epithelioid histology, who achieved an OS of 12.3 with pembrolizumab added to chemotherapy versus 8.2 months with chemotherapy. Patients with epithelioid histology had an OS of 19.8 with chemo-immunotherapy compared to an OS of 18.2 months with chemotherapy alone. Despite the greater difference in survival for patients with non-epithelioid tumors, in this study, unlike in CheckMate-743, the magnitude of the overall effect of immunotherapy was greater for patients with epithelioid histology, who showed an OS of 19.8 months vs. 12.3 months for patients with non-epithelioid tumors.

Initial second-line and subsequent immunotherapy studies that evaluated the efficacy of treatment according to histology reported a greater efficacy of immunotherapy for treating non-epithelioid tumors in trials with nivolumab (NivoMes and MERIT) and durvalumab plus tremelimumab (MESO-TREM) [18,19,20,21,22,23]. However, in second-line studies with a larger number of patients, greater efficacy for immunotherapy among patients with non-epithelioid histology has not been demonstrated. The CONFIRM trial reported a significant improvement in PFS and OS with nivolumab in the epithelioid group [23]. Similarly, in the PROMISE-meso trial with pembrolizumab, patients with tumors with non-epithelioid histology showed worse PFS and OS, although these data were not statistically significant, probably due to the limited sample size [20].

### 4.3. TMB

TMB is considered a potential predictive biomarker of response to immunotherapy for some tumors such as lung cancer and melanoma [39,40]. In KEYNOTE-158, a response rate advantage was found in the high-TMB group [36]. However, considering the mesothelioma cohort, in 85 evaluable cases, only 1 patient had high TMB, and a response was reported in 9 of 84 patients with low TMB. The same tissue TMB score was observed for both responders and non-responders to pembrolizumab (1.26 mutations per megabase). An exploratory analysis regarding TMB was also performed in the Checkmate-743 trial [15]. TMB assessment was feasible for 53% of patients treated with nivolumab plus ipilimumab and for 45% of patients treated with the chemotherapy arm, with a low mean TMB value (1.75 mut/Mb). In this analysis, higher mutational burden also did not correlate with better survival in the immunotherapy or chemotherapy arms.

### 4.4. Crhomosomics Rearrengements

Chromothripsis has been associated with a worse prognosis among patients with MPM [41,42]. However, this structural chromosomal variant is associated with the potential formation of neoantigens that facilitate the intratumoral expansion of T cell clones, suggesting that chromothripsis could play a role in the response to immunotherapy. In a retrospective study that explored the correlation between chromosomal rearrangements and survival among patients with MPM who received nivolumab or ipilimumab in combination with nivolumab, it was found that rearrangements were not predictors of efficacy, but genetic signatures associated with presentation and antigen processing predicted an OS difference of more than 1.5 years [43]. Further work will be required to demonstrate whether these chromosomal rearrangements may present an opportunity for a biomarker of response to immune checkpoint inhibitors.

### 4.5. Genomic Markers

To date, the most in-depth analysis of genomic and phenotypic factors correlated with immunotherapy outcome was performed on patients from the PrE0505 study treated with chemotherapy plus durvalumab [32]. Chromosomal instability was identified to occur more frequently in epithelioid MPM, with an OS of more than 12 months. The authors also demonstrated that a higher burden of immunogenic mutations in major histocompatibility complex (MHC) class I and MHC class II was significantly associated with a better response to durvalumab plus chemotherapy (*p* = 0.064 and *p* = 0.023, respectively), especially in the epithelioid group. Furthermore, better survival was observed among patients with high variability in T cell receptor (TCR) clonality and worse survival in the APOBEC signature. The authors demonstrated that greater divergence of the human leukocyte antigen (HLA)-B locus was related to a better radiological response to chemo-immunotherapy, particularly in relation to epithelioid MPM. Despite the limitations that come with being a small cohort study, this analysis revealed a subset of genomic and immunological characteristics that could predict outcomes after chemo-immunotherapy for MPM.

*BAP1* is a tumor suppressor gene that represents the most commonly mutated gene in MPM, especially in epithelioid tumors [44]. The PrE0505 trial with durvalumab plus chemotherapy showed that germline mutations in *BAP1* were associated with significantly prolonged survival after chemo-immunotherapy [32].

The second most common somatic mutation in MPM patients is the 9p21 deletion, which contains *CDK2N2A*. A pan-cancer analysis of data from The Cancer Genome Atlas (TCGA) involving eight immunotherapy trials, which did not include MPM patients, showed that loss of 9p21 is associated with a “cold” tumor microenvironment [45]. Considering that almost 50% of TCGA samples in the MPM cohort present loss of 9p21, this mechanism represents an important explanation for immunotherapy resistance in MPM [46].

The Checkmate-743 study showed that a high four-gene inflammatory signature score (*CD8A*, *STAT1*, *LAG3*, and *CD274*) was associated with improved OS in the nivolumab plus ipilimumab arm (21.8 versus 16.8 months among patients with low scores) [29]. In the chemotherapy arm, no correlation was identified between the inflammatory gene signature score and response to treatment. The inflammatory signature score could, therefore, be considered a positive predictive biomarker of response to immunotherapy.

### 4.6. Time

It has been suggested that the type of TIME (tumor immune microenvironment) may affect the outcome of immunotherapy [47,48,49,50]. In MPM, a comprehensive immunoproteogenomic analysis defined two disparate TIMEs: TIME-I, which is characterized by a greater number of PD-1 + CTLA-4 + CD8 + T cells, and the TIME-II subtype, which contains more Tregs and naïve CD8+ T cells. TIME-I was reported to be associated with a better response to immunotherapy [51]. In another study, TIME was classified according to NanoString analysis into group 1 with poor gene expression associated with the immune system; group 2 with moderate T cell effector gene expression and a high level of B cell gene expression; and group 3, which had high expression of PD-L1 and T cell effector gene [52]. These comprehensive studies indicate the potential predictive value of TIME for MPM immunotherapy, but this hypothesis has not been verified in clinical trials with immunotherapy.

Preliminary data presented in the DREAM study have indicated that CD8 density within tumor biopsies does not predict response, but it was significantly correlated with better PFS and OS when CD8 was quantified only within epithelial areas [32].

## 5. New Immunotherapy Drugs under Investigation

### 5.1. Therapies Based on Mesothelin

Mesothelin is a glycoprotein that is expressed in many solid tumors, including MPM. Different drugs targeting mesothelin inhibition have been explored in previously treated MPM patients.

CRS-207 is a live, attenuated, non-virulent strain of Listeria Monocytogenes that encodes human mesothelin. A phase Ib clinical trial found complete responses in 3% of patients and partial responses in 54% of patients [53]. However, a subsequent phase II study did not show clinical activity of the combination of CRS-207 with PD-1 inhibition in the interim analysis, so clinical development of this therapy was discontinued.

The study with the largest number of patients with mesothelin inhibitors published to date did not demonstrate efficacy against chemotherapy [54]. Anetumab ravtansine, an anti-mesothelin antibody conjugate linked to the tubulin inhibitor DM4, was evaluated in a phase II trial, wherein patients were randomized to receive anetumab ravtansine or vinorelbine. No differences were found in PFS or OS (4.3 versus 4.5 months and 10.1 versus 11.6 months for anetumab ravtansine and vinorelbine, respectively).

### 5.2. Checkpoint Inhibitors

In preclinical models, the combination of anti-LAG-3 and anti-PD1 reduced tumor size and provided a survival benefit, and a recent phase I study using tebotelimab found a partial response in a patient with MPM [55,56]. There is currently an ongoing clinical trial with Ieramilimab in combination with the anti-PD1 PDR001 for patients with advanced tumors, including a cohort of patients with MPM.

VISTA is an immune checkpoint gene that has strong expression in epithelioid MPM, above the levels of other solid tumors [5,57,58]. A phase I clinical trial with CA-170, a small-molecule inhibitor of VISTA, which included a cohort of 12 patients with MPM, did not show partial or complete responses in any of the patients. However, to date, VISTA has not been investigated as a potential predictive biomarker of response to immunotherapy.

Due to the high expression of B7-H3 in patients with MPM, this molecule has been proposed as a potential therapeutic target [59]. In a series of 44 MPM samples, B7-H3 was found to be expressed in 41 of the 44 patients. In another study considering histology, it was confirmed that B7-H3 was expressed in almost all patients (90% of epithelioid tumors and 89% of non-epithelioid tumors) [60]. Regarding its prognostic value, a TCGA analysis showed that B7-H3 expression was associated with worse survival.

Several studies have shown that TIM-3 expression was high in tumor-infiltrating lymphocytes (TILs) in different tumors, including MPM [61]. A study involving 54 patients with MPM found that TIM-3 expression occurred in up to 40% of patients and correlates with PD-L1 [62]. A phase I study including MPM patients treated with the monoclonal antibody INCAGN02390 targeting TIM-3 is currently underway.

### 5.3. Vaccines

Dendritic cell (DC) therapy is intended to induce the proliferation of T cells and promote the activation of CD4+ and CD8+ T-cells by presenting them with tumor antigens, allowing CD8+ T-cells to infiltrate the tumor microenvironment [63]. In mesothelioma, clinical research with dendritic cells has shown notable antitumor activity in two studies with a small number of patients, with a survival of up to 24 months among some patients [64,65]. To validate these promising results, a European randomized phase II/III trial (DENIM study) evaluating dendritic cell immunotherapy as a maintenance treatment after standard first-line chemotherapy is underway (Table 3).

Other immunotherapeutic strategies based on the local delivery of cytokines through genetically modified viruses have been evaluated. Adenovirus-delivered interferon alfa-2b (Ad-IFN) is a replication-defective adenoviral vector containing the human interferon-alpha2b gene. Intrapleural administration of Ad-IFN in combination with celecoxib and chemotherapy was evaluated in a cohort of 40 patients, achieving a response rate of 25% and a survival value of 21.5 months [66]. Following these promising results, the phase III INFINITE trial was launched, which is evaluating the efficacy of intrapleural Ad-IFN in combination with celecoxib and gemcitabine in the second- or third-line context, and the results are expected in 2024. ONCOS-102 is a GM-CSF oncolytic adenovirus that demonstrated activity in a phase 1–2 study in combination with cisplatin and pemetrexed [67]. Phase 1 included 15 patients, and it was shown that the administration of the virus was safe and induced immune activation. Analysis of tumor tissue revealed ONCOS-102-induced modulation of the tumor microenvironment with an increased population of cytotoxic T cells. Furthermore, immune activation was associated with tumor responses and was more pronounced in patients with better survival outcomes. Other phase 1 trials with oncolytic viruses have shown preliminary efficacy [68,69]. Ad-SGE-REIC is an adenoviral vector carrying REIC/Dkk that was administered via an intrapleural route in a series of 13 patients. The treatment did not demonstrate an objective response, and the median PFS was 3.4 months [68]. Another phase 1 trial involving HSV1716, oncolytic herpes simplex virus, injected via the intrapleural route did not demonstrate clinical responses, but disease stabilization was reported in 50% of the patients [69].

In the second line of treatment, the NIPU study evaluated the addition of the UV1 anti-telomerase vaccine in combination with ipilimumab and nivolumab [70]. The study did not demonstrate improvement in the primary endpoint of PFS for patients who received the UV1 vaccine, but it did find an improvement in survival (15.4 months vs. 11.1 months for patients who received UV1 vs. those who did not).

### 5.4. CAR-T

CAR-Ts were investigated in two phase I clinical trials that showed moderate clinical responses [71,72]. A third phase I clinical study was designed to investigate the safety and feasibility of CART-mesothelin cells administered intravenously or intrapleurally with or without cyclophosphamide in solid tumors, including epithelioid MPM. This study is ongoing, and data are not yet available. A fourth phase 1 trial explored the intrapleural administration of anti-FAP CART-T cells and demonstrated that it was well tolerated and that CART-T cells were detected in the peripheral blood [73]. Finally, the intrapleural administration of CAR-T targeting mesothelin followed by the administration of pembrolizumab has been explored, finding a median survival of 23.9 months among previously treated patients [74]. Although the response rate found was lower than expected (two partial responses out of 16 treated patients), these responses were maintained for more than six months.

## 6. Discussion

MPM is characterized as an aggressive tumor related to exposure to asbestos with a decreasing incidence in recent years in developed countries. Genomic alterations in MPM are not as frequent as in other tumors. The mutational landscape is dominated by the loss of function of suppressor genes that lack a therapeutic target [4]. However, the chronic inflammatory response generated by asbestos creates a tumor microenvironment equivalent to that of an immunologically active tumor.

Several factors, such as clinical stage, histology, performance status, and comorbidities, should be considered when designing therapeutic strategies for patients with MPM. Considering that the systemic treatment for MPM is not curative, maintaining the quality of life of the patients should be a priority. The role of surgery is crucial for diagnosis, the staging of patients, and the palliation of symptoms. Regarding its role in the treatment of patients with early-stage MPM, selective surgery is recommended for select patients and in select centers with experience and as part of multimodality treatment.

Systemic treatment is recommended as part of a multimodal treatment for early stages and is the main treatment for patients with advanced disease. For two decades, the only systemic treatment for patients with MPM has been chemotherapy with cisplatin and pemetrexed, achieving a median survival of 12 months [2]. For patients who progress to platinum–pemetrexed combination treatment, there are no other approved chemotherapy options. However, in recent years, there have been significant improvements in our knowledge of the biology of mesothelioma. The greatest advance has been made in the field of immunotherapy, which has improved the survival of patients with different types of cancer and now also mesothelioma. The combination of two immunotherapy agents with nivolumab plus ipilimumab in the CheckMate-743 study demonstrated an increase in survival, positioning the immunotherapy combination as a valid alternative to first-line chemotherapy for patients with MPM [3]. In this study, the magnitude of the benefit of immunotherapy was greater for patients with PD-L1 expression greater than 1% and for patients with non-epithelioid histology.

Another first-line treatment option is the combination of chemotherapy with immunotherapy. To date, two single-arm phase II studies have reported an interesting benefit from the combination of durvalumab with chemotherapy (DREAM and PRE0505), while the results of a randomized phase III study are pending [31,32]. In 2023, the results of the first randomized trial that evaluated the role of chemo-immunotherapy versus chemotherapy in the first line of treatment were presented [34]. The IND227 study showed that the combination treatment achieved a survival of 17.3 months compared to that of 16.1 months with chemotherapy. In this study, as in CheckMate-743, better results were found for patients with non-epithelioid histology due to the lower efficacy of chemotherapy in treating non-epithelioid tumors, but, in this case, the overall benefit of immunotherapy was better for patients with epithelioid tumors (OS of 19.8 months vs. 12.3 months for patients with epithelioid and non-epithelioid tumors treated with pembrolizumab). In this trial, no significant differences were found with regard to the level of PD-L1 expression. Currently, another phase III trial involving chemo-immunotherapy (BEAT-MESO) has completed its recruitment stage and is awaiting results.

With the current results of the first-line studies, we are faced with three treatment options: chemotherapy, the combination of immunotherapy with nivolumab plus ipilimumab, or chemotherapy plus immunotherapy (Figure 1). Based on what has been reported in immunotherapy studies, the results are superior with respect to chemotherapy, so the option of first-line chemotherapy should be reserved for patients with contraindications for receiving immunotherapy treatment or patients with a performance status of 2. Regarding the best choice of first-line treatment considering the options of nivolumab plus ipilimumab versus chemo-immunotherapy, at the moment, there are no studies that answer this question. On the one hand, the combination of chemotherapy plus immunotherapy releases neoantigens that could induce a better response with immunotherapy, but on the other hand, the addition of chemotherapy significantly increases toxicity. The EVOLVE clinical trial exploring the combination of anti-CTLA4 therapy with anti-PD-1 plus chemotherapy versus chemotherapy alone or the combination of nivolumab plus ipilimumab could answer whether chemo-immunotherapy is superior to the immunotherapy doublet, although this study will use double CTLA-4 and PD-1 blockade as a supplement to chemotherapy.

The identification of predictive biomarkers of response to immunotherapy that can help us select patients with the greatest possibility of benefiting from treatment is an unmet need. Compared with that for other cancers, biomarker research has made limited progress. Although the predictive role of PD-L1 is well established for different tumors, its role in MPM is more controversial.

Histology has been postulated as a potential predictor of response to immunotherapy. In the Chekmate-743 trial and IND227, a greater benefit of immunotherapy was reported for patients with non-epithelioid histology [15,34]. It should be noted that in these studies, the magnitude of the benefit with immunotherapy was similar in both histologies in the nivolumab plus ipilimumab treatment, but the efficacy of pembrolizumab was clearly lower in non-epithelioid compared to epithelioid tumors. The randomized CONFIRM study with nivolumab demonstrated opposite results, with better survival among patients with epithelioid histology [23]. Analysis of future randomized immunotherapy trials will be necessary to discern the possible predictive role of histology.

Regarding other biomarkers, the TMB in patients with MPM is low, and, so far, no greater efficacy has been demonstrated with immunotherapy among patients with a higher TMB. Promising results were found with the inflammatory gene signature in the Checkmate-743 study [15].

## 7. Conclusions

In conclusion, fortunately, the growing research in the field of MPM has culminated in the approval of new therapies that improve patient survival. However, new combinations of immunotherapy with chemotherapy could further improve patient outcomes. Although MPM is a disease with a globally decreasing incidence, there is still a long way to go to optimize the first line of treatment and approve therapies after the progression of immunotherapy.

## Figures and Tables

**Figure 1 cancers-15-05787-f001:**
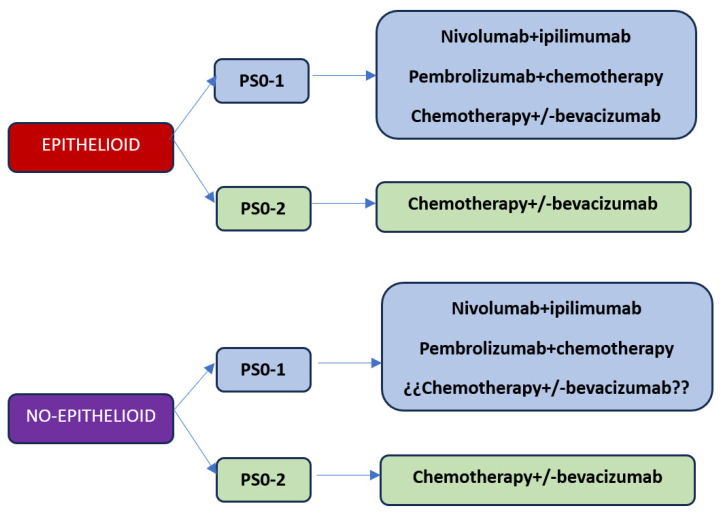
MPM treatment proposal for the year 2024. Figure legend: Proposed first-line treatment algorithm for patients with MPM. Treatment selection according to histology and PS (performance status).

**Table 1 cancers-15-05787-t001:** Summary of immunotherapy studies in MPM.

Trial	Drug	N	PD-L1 Selected	Response Rate (%)	PFS (Months)	OS (Months)
Keynote028	Pembrolizumab	25	Yes	20	5.4	18
Keynote158	Pembrolizumab	118	No	8	2.1	10
PROMISE	Pembrolizumab	73	No	22	2.5	10.7
NivoMes	Nivolumab	34	No	15	3.6	11.8
Javelin	Avelumab	53	No	8	4.1	10.7
MERIT	Nivolumab	34	No	29	6.1	17.3
CONFIRM	Nivolumab	332	No	11	3	9.2
NIBITMESO	Durvalumab–Tremelumumab	40	No	28	5.7	16.5
INITIATE	Nivolumab–ipilimumab	38	No	30	6.2	64% at 12 months
MAPS	Nivolumab–ipilimumab	125	No	52	5.6	15.9
CheckMate-743	Nivolumab–ipilimumab	303	No	40	6.8	18.1
IND227	Pembrolizuma–chemotherapy	222	No	63	7.1	17.3
DREAM	Durvalumab–chemotherapy	54	No	61	6.9	18.4
PrE0505	Durvalumab–chemotherapy	55	No	56.4	6.7	20.4
JME-01	Nivolumab–chemotherapy	18	No	77	8	20.8

PFS: progression-free survival, OS: overall survival.

**Table 2 cancers-15-05787-t002:** Predictive factors of response to immunotherapy.

Factor	Predictive Value	Comments
PD-L1	Unclear	Pembrolizumab and nivolumab are not predictive (PROMISE, CONFIRM) Nivolumab yields better OS in PD-L1 + (CHECKMATE743) Chemo + immunotherapy is not predictive (DREAM)
Histology	Unclear	Nivolumab + ipilimumab and pembrolizumab + chemo improve OS for no-epithelioid patients (CHECKMATE743,IND227) Nivolumab 2nd line offers better outcomes for epithelioid patients (CONFIRM) Pembrolizumab 2nd line better outcomes for epithelioid patients (PROMISE)
TMB	Limited data	Nivolumab–ipilimumab exhibited no correlation (CHECKMATE743)
Chromosomal rearrangements	Limited data	Nivolumab + ipilimumab generates signature predictive OS (CHECKMATE743)
Genomic markers	Limited data	MHC, TCR, APOBEC, and germline mutation BAP1 yield predictive outcomes durvalumab + chemo (PRE0505) 4-gene inflammatory signature better OS with Nivolumab–ipilimumab (CHECKMATE743)
TIME	Limited data	PD1, CTLA4, CD8, and gene expression patterns are associated with outcomes in precilinical data

OS: overall survival, TMB: tumor mutation burden, and TIME: tumor immune microenvironment.

**Table 3 cancers-15-05787-t003:** Selected completed or ongoing trials with vaccines.

Trial Code	Treatment	Phase	Status
NCT00280982	Tumor-lysate-loaded autologous dendritic cells	I	Completed
NCT10241682	Dendritic cells + chemotherapy	I	Completed
NCT03610360 (DENIM)	Dendritic cells after chemotherapy	II/III	Recruiting
NCT02649829 (MESODEC)	Autologous dendritic cells	I/II	Active, non recruiting
NCT03546426 (MESOVAX)	Pembrolizumab plus autologous dendritic cells	I	Recruiting
NCT01265433	WT-1 analog + GM-CSF	II	Completed
NCT04040231	WT-1 analog + nivolumab	I	Active, non-recruiting
NCT05765084 (Immuno-MESODEC)	Atezolizumab and WT1/DC vaccination	I/II	Recruiting
NCT01675765	Listeria-monocytogenes-expressing mesothelina (CRS-207) + CT	I	Completed
NCT01119664	rAd-IFNa2B + chemotherapy	I/II	Completed
NCT03710876 (INFINITE)	rAd-IFN + celecoxib+ gemcitabine	III	Active, non-recruiting
NCT01503177	Intrapleural measles virus	I	Completed
NCT06031636	Oncolytic adenovirus (H101) + PD-1 inhibitors	Observational	Recruiting
NCT01721018	Intrapleural HSV1716	1/2	Completed
NCT02879669	ONCOS-102 with chemotherapy	I/II	Completed
NCT04013334	MTG201 (Ad-SGE-REIC/Dkk-3) + nivolumab	II	Active no recruiting

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
