# Peer review of "Current State-of-the-Art Therapy for Malignant Pleural Mesothelioma and Future Options Centered on Immunotherapy"

_cancers, 2023, doi:10.3390/cancers15245787_

Round 1
Reviewer 1 Report
Comments and Suggestions for Authors
Please provide a list of abbreviations.
The section 4.6 TIME requires additional referencing
A table summarizing the passive and active immunotherapeutic approaches should be included in the vaccines section.
Clinical trials testing the efficiency of oncolytic therapeutic vaccines should be included/expanded in the vaccines section.
Please provide a figure legend for the figure 1.
Comments on the Quality of English LanguageMinor grammatical errors: “…did not demonstrated differences…”
Some sentences need to be rewritten for clarity:
"...but PFS and OS were significantly better when CD8 was quantified..."
"Genomic alterations in MPM are not as frequent as in other tumors, being dominated by the loss of suppressor genes that lack a therapeutic target"
Author Response
Reviewer 1
Please provide a list of abbreviations.
Following your recommendation we have added a list of abbreviations at the end of the manuscript
CTLA4: cytotoxic T lymphocyte antigen 4
ECOG PS: Eastern Cooperative Oncology Group Performance Status
EMA: European Medicines Agency
FDA: food and drug administration
HLA: human leukocyte antigen
LIPI: pulmonary immune prognostic index
PFS: progression free survival
ORR: overall response rate
OS: overall survival
PD-L1 programmed cell death ligand 1
TCR T cell receptor
TIM3: T-cell immunoglobulin mucin 3
TTField: alternating electric field therapy
VISTA: V-domain Ig-containing suppressor of T cell activation
The section 4.6 TIME requires additional referencing
Thank you for your comment, following your recommendation we have added some references
Hugo W, Zaretsky JM, Sun L, Song C, Moreno BH, Hu-Lieskovan S, et al. Genomic and transcriptomic features of response to anti-PD-1 therapy in metastatic melanoma. Cell (2016) 165(1):35–44. doi: 10.1016/j.cell.2016.02.065
Jiang P, Gu S, Pan D, Fu J, Sahu A, et al. Signatures of T cell dysfunction and exclusion predict cancer immunotherapy response. Nat Med. 2018 Oct;24(10):1550-1558.
Chee SJ, Lopez M, Mellows T, Gankande S, Moutasim KA, Harris S, et al. Evaluating the effect of immune cells on the outcome of patients with mesothelioma. Br J Cancer (2017) 117(9):1341–8. doi: 10.1038/bjc.2017.269
Mannarino L, Paracchini L, Pezzuto F, Olteanu GE, Moracci L, et al. Epithelioid Pleural Mesothelioma Is Characterized by Tertiary Lymphoid Structures in Long Survivors: Results from the MATCH Study. Int J Mol Sci. 2022 May 21;23(10):5786.
A table summarizing the passive and active immunotherapeutic approaches should be included in the vaccines section.
We have added a table with the information requested.
Trial code |
Treatment |
Phase |
Status |
NCT00280982 |
Tumor lysate loaded autologous dendritic cells |
I |
Completed |
NCT10241682 |
Dendritic cells + chemotherapy |
I |
Completed |
NCT03610360 (DENIM) |
Dendritic cells after chemotherapy |
II/III |
Recruiting |
NCT02649829 (MESODEC) |
Autologous Dendritic Cell
|
I/II |
Active, non recruiting |
NCT03546426 (MESOVAX) |
Pembrolizumab Plus Autologous Dendritic Cell
|
I |
Recruiting |
NCT01265433 |
WT-1 analog + GM-CSF |
II |
Completed
|
NCT04040231 |
WT-1 analog + nivolumab
|
I |
Active, non recruiting |
NCT05765084 (Immuno-MESODEC) |
Atezolizumab and WT1/​DC Vaccination
|
I/II |
Recruiting |
NCT01675765 |
Listeria monocytogenes expressing mesothelina (CRS-207) + CT |
I |
Completed |
NCT01119664 |
rAd-IFNa2B + Chemotherapy |
I/II |
Completed |
NCT03710876 (INFINITE) |
rAd-IFN + celecoxib+ gemcitabine |
III |
Active, non recruiting |
NCT01503177 |
Intrapleural Measles Virus |
I |
Completed |
NCT06031636 |
Oncolytic Adenovirus(H101) + PD-1 Inhibitors |
Observational |
Recruiting |
NCT01721018 |
Intrapleural HSV1716 |
1/2 |
Completed |
NCT02879669 |
ONCOS-102 with chemotherapy |
I/II |
Completed |
NCT04013334 |
MTG201 (Ad-SGE-REIC/Dkk-3) + nivolumab |
II |
Active no recruiting |
Clinical trials testing the efficiency of oncolytic therapeutic vaccines should be included/expanded in the vaccines section.
We appreciate the reviewer’s comments. We have included more information regarding vaccines in the Table 3 and also a comment in the manuscript.
Other immunotherapeutic strategies based on the local delivery of cytokines through genetically modified viruses have been evaluated. Adenovirus-delivered interferon alfa-2b (Ad-IFN) is a replication-defective adenoviral vector containing the human interferon-alpha2b gene. Intrapleural administration of Ad-IFN in combination with celecoxib and chemotherapy was evaluated in a cohort of 40 patients, achieving a response rate of 25% and survival of 21.5 months [66]. Following these promising results, the phase III INFINITE trial was launched, which evaluates the efficacy of intrapleural Ad-IFN in combination with celecoxib and gemcitabine in the second or third line context and the results are expected in 2024. ONCOS-102 is a GM-CSF oncolytic adenovirus that demonstrated activity in a phase 1-2 study in combination with cisplatin and pemetrexed [67]. The phase 1 included 15 patients and it was shown that the administration of the virus was safe and produced immune activation. Analyzes of tumor tissue revealed ONCOS-102-induced modulation of the tumor microenvironment with an increased population of cytotoxic T cells. Furthermore, immune activation was associated with tumor responses and was more pronounced in patients with better survival outcomes. Other phase 1 trials with oncolytic virus have shown preliminary efficacy [68,69]. Ad-SGE-REIC is an adenoviral vector carrying REIC/Dkk administrated intrapleural in a series of 13 patients. The treatment did not demonstrate objective response and the median PFS was 3.4 months [68]. Another phase 1 with HSV1716, oncolytic herpes simplex virus, injected via intrapleural did not demonstrate clinical responses but disease stabilization was reported in 50% of the patients [69].
Please provide a figure legend for the figure 1.
By mistake the legend was not included in the article. We have added the legend.
Figure legend: Proposed first-line treatment algorithm for patients with MPM. Treatment selection according to histology and PS (performance status)
Comments on the Quality of English Language
We have reviewed the manuscript and corrected the mistakes
Minor grammatical errors: “…did not demonstrated differences…”
Some sentences need to be rewritten for clarity:
"...but PFS and OS were significantly better when CD8 was quantified..." Preliminary data presented in the DREAM study have indicated that CD8 density within tumor biopsies does not predict response, but it was significantly correlated with better PFS and OS when CD8 was quantified only within epithelial areas [32].
"Genomic alterations in MPM are not as frequent as in other tumors, being dominated by the loss of suppressor genes that lack a therapeutic target"
Genomic alterations in MPM are not as frequent as in other tumors. The mutational landscape is dominated by the loss of function in suppressor genes that lack a therapeutic target"

Reviewer 2 Report
Comments and Suggestions for Authors
Firstly, I want to thank the authors for the opportunity to read this very nice review about the current state of art therapy for pleural mesothelioma patients. This review provides a great overview about the latest important clinical trials in the therapy of MPM.
I have a few minor comments:
1. Paragraph 2.1: Treatment surgery: As the official paper for the MARS2 trial is not published yet, I would be cautious with stating or repeating the PIs (Dr.Eric Lim) conclusion. There are numerous mistakes in the design as no standardized staging methods as EBUS etc. No volumetry pre and post chemotherapy for the evaluation of therapy response according to mRECIST criteria. And so on. Either this is going to be discussed and mentioned among other points or this should be deleted and only mentioned as results have been presented and highly discussed already among the specialists.
2. I would suggest to create a table of all the mentioned trials in regard to their usage (first/second line etc). Additionally; I would suggest a table with an overview of all predictive factors (histology, PD-L1 etc) discussed in this paper.
3. Two abbreviations are not spelled out: TMB and TIME. I would write out these abbreviations once
4. I miss a reference regarding CAR-T cell therapy by the team around Curioni and Petrausch, both well known MPM specialists. https://doi.org/10.1093/annonc/mdz253.0525
5. The discussion should elaborate the different treatment therapies according to the tumor stage as well as mention the different therapy approaches. Multimodal therapy, Only systemic therapy and best supportive care. This would give the discussion a better structure and it is not only a repetition of the sections before.
Overall, this is a nice overview. Congratulations to the authors.
Author Response
I have a few minor comments:
Paragraph 2.1: Treatment surgery: As the official paper for the MARS2 trial is not published yet, I would be cautious with stating or repeating the PIs (Dr.Eric Lim) conclusion. There are numerous mistakes in the design as no standardized staging methods as EBUS etc. No volumetry pre and post chemotherapy for the evaluation of therapy response according to mRECIST criteria. And so on. Either this is going to be discussed and mentioned among other points or this should be deleted and only mentioned as results have been presented and highly discussed already among the specialists.
We absolutely agree with reviewer´s comment and following your recommendation we have made some changes
Most patients with mesothelioma are not candidates for surgery due to the extent of the disease and comorbidities. The MARS study, which included 50 patients, demonstrated that extrapleural pneumonectomy was associated with severe complications and did not improve survival compared to patients who did not undergo surgery [12]. In 2023, the results of the MARS-2 study were presented comparing the treatment of chemotherapy and pleuro-decortication versus chemotherapy without surgery [13]. The study found that survival was superior for patients who did not receive surgery. However, this study has not yet been published and the results should be interpreted with caution. More patients with tumors spreading to the lungs were included in the surgery arm. In addition, the preoperative staging of patients was not standardized and no volumetry pre and post chemotherapy for the evaluation of therapy response according to modified RECIST criteria. These issues need to be clarified to establish the role of surgery in MPM.
I would suggest to create a table of all the mentioned trials in regard to their usage (first/second line etc). Additionally; I would suggest a table with an overview of all predictive factors (histology, PD-L1 etc) discussed in this paper.
Following reviewer´s comment we have included all the trials in a table and included a new table with the predictive factors
Table 1 summary of immunotherapy studies in MPM.
Trial |
Drug |
N |
PD-L1 selected |
Response rate (%) |
PFS (months) |
OS (months) |
Keynote028 |
Pembrolizumab |
25 |
Yes |
20 |
5.4 |
18 |
Keynote158 |
Pembrolizumab |
118 |
No |
8 |
2.1 |
10 |
PROMISE |
Pembrolizumab |
73 |
No |
22 |
2.5 |
10.7 |
NivoMes |
Nivolumab |
34 |
No |
15 |
3.6 |
11.8 |
Javelin |
Avelumab |
53 |
No |
8 |
4.1 |
10.7 |
MERIT |
Nivolumab |
34 |
No |
29 |
6.1 |
17.3 |
CONFIRM |
Nivolumab |
332 |
No |
11 |
3 |
9.2 |
NIBITMESO |
Durvalumab-tremelumumab |
40 |
No |
28 |
5.7 |
16.5 |
INITIATE |
Nivolumab-ipilimumab |
38 |
No |
30 |
6.2 |
64% at 12 months |
MAPS |
Nivolumab-ipilimumab |
125 |
No |
52 |
5.6 |
15.9 |
CheckMate-743 |
Nivolumab-ipilimumab |
303 |
No |
40 |
6.8 |
18.1 |
IND227 |
Pembrolizumab-chemotherapy |
222 |
No |
63 |
7.1 |
17.3 |
DREAM |
Durvalumab-chemotherapy |
54 |
No |
61 |
6.9 |
18.4 |
PrE0505 |
Durvalumab-chemotherapy |
55 |
No |
56.4 |
6.7 |
20.4 |
JME-01 |
Nivolumab-chemotherapy |
18 |
No |
77 |
8 |
20.8 |
Table 2: predictive factors of response to immunotherapy
Factor |
Predictive value |
Comments |
PD-L1 |
Unclear |
Pembrolizumab and nivolumab not predictive (PROMISE, CONFIRM) Nivolumab better OS PD-L1+ (CHECKMATE743) Chemo+immunotherapy not predictive (DREAM) |
Histology |
Unclear |
Nivolumab+ipilimumab and pembrolizumab+chemo improve OS no-epithelioid (CHECKMATE743,IND227) Nivolumab 2nd line better outcomes epithelioid (CONFIRM) Pembrolizumab 2nd line better outcomes epithelioid (PROMISE) |
TMB |
Limited data |
Nivolumab-ipilimumab no correlation (CHECKMATE743) |
Chromosomal rearrangements |
Limited data |
Nivolumab+ipilimumab generates signature predictive OS (CHECKMATE743) |
Genomic markers |
Limited data |
MHC, TCR, APOBEC and germline mutation BAP1 predictive outcomes durvalumab+chemo (PRE0505) 4-gene inflammatory signature better OS nivolumab-ipilimumab (CHECKMATE743) |
TIME |
Limited data |
PD1, CTLA4, CD8 and gene expression patterns associated outcomes in precilinical data |
OS: overall survival, TMB: tumor mutation burden, TIME: tumor immune microenvironment
Two abbreviations are not spelled out: TMB and TIME. I would write out these abbreviations once
Thank your for your observation. We have corrected the mistake
- I miss a reference regarding CAR-T cell therapy by the team around Curioni and Petrausch, both well known MPM specialists. https://doi.org/10.1093/annonc/mdz253.0525
We have added this reference
A fourth phase 1 explored the intrapleural administration of anti-FAP CART-T cells and demonstrated that was well tolerated and CART-T cells were detected in the peripheral blood [73]
- Curionim C Britshgi, S Hiltbrunner, et al. A phase I clinical trial of malignant pleural mesothelioma treated with locally delivered autologous anti-FAP-targeted CAR t-cells. Annals of Oncology 2019, Vol 39, supp 5, v501.
The discussion should elaborate the different treatment therapies according to the tumor stage as well as mention the different therapy approaches. Multimodal therapy, Only systemic therapy and best supportive care. This would give the discussion a better structure and it is not only a repetition of the sections before.
Following your recommendation we have added some information in the discussion previous to systemic treatment section.
Several factors such as clinical stage, histology, performance status and comorbidities should be considered to design the therapeutic strategy of patients with MPM. Considering that the systemic treatment for MPM is not curative, maintaining the quality of life of the patients should be a priority. The role of surgery is crucial for diagnosis and to stage the patients and for palliation of symptoms. Regarding its role for treatment of patients with early stage, selective surgery is recommended in selected patients and in selected centers with experience and as part of multimodality treatment.
Systemic treatment is recommended as part of a multimodal treatment for early stages and the only treatment for patients with advanced disease.
